# Linear Relationships between Partition Coefficients of Different Organic Compounds and Proteins in Aqueous Two-Phase Systems of Various Polymer and Ionic Compositions

**DOI:** 10.3390/polym12071452

**Published:** 2020-06-29

**Authors:** Nuno R. da Silva, Luisa A. Ferreira, Pedro P. Madeira, José A. Teixeira, Vladimir N. Uversky, Boris Y. Zaslavsky

**Affiliations:** 1IBB—Institute for Biotechnology and Bioengineering, Centre of Biological Engineering, Universidade do Minho, Campus de Gualtar, 4710-057 Braga, Portugal; nuno.silva@ceb.uminho.pt (N.R.d.S.); jateixeira@deb.uminho.pt (J.A.T.); 2Cleveland Daignostics, 3615 Superior Ave., Cleveland, OH 44114, USA; Luisa.Ferreira@cleveland-diagnostics.com; 3Centro de Investigacao em Materiais Ceramicos e Compositos, Department of Chemistry, University of Aveiro, 3810-193 Aveiro, Portugal; p.madeira@ua.pt; 4Department of Molecular Medicine, Morsani College of Medicine, University of South Florida, Tampa, FL 33612, USA; 5Laboratory of New Methods in Biology, Institute for Biological Instrumentation, Russian Academy of Sciences, Federal Research Center “Pushchino Scientific Center for Biological Research of the Russian Academy of Sciences”, 142290 Pushchino, Russia

**Keywords:** aqueous two-phase systems, partitioning, organic compounds, drugs, proteins, blood–tissue distribution

## Abstract

Analysis of the partition coefficients of small organic compounds and proteins in different aqueous two-phase systems under widely varied ionic compositions shows that logarithms of partition coefficients for any three compounds or proteins or two organic compounds and one protein are linearly interrelated, although for protein(s) there are ionic compositions when the linear fit does not hold. It is suggested that the established interrelationships are due to cooperativity of different types of solute–solvent interactions in aqueous media. This assumption is confirmed by analysis of distribution coefficients of various drugs in octanol-buffer systems with varied ionic compositions of the buffer. Analysis of the partition coefficients characterizing distribution of variety of drugs between blood and different tissues of rats in vivo reported in the literature showed that the above assumption is correct and enabled us to identify the tissues with the components of which the drug(s) may engage in presumably direct interactions. It shows that the suggested assumption is valid for even complex biological systems.

## 1. Introduction

It has been shown [1] that for three different solutes (such as organic compounds, salts, and polymers), different physicochemical properties of their aqueous solutions (such as water activity, osmotic coefficient, relative permittivity, viscosity, and surface tension) are linearly related over a relatively wide range of solute concentrations and may be described as:Y^i^_1_(c^i^_1_) = k_1_ + k_2_Y^i^_2_(c^i^_2_) + k_3_Y^i^_3_(c^i^_3_),(1)
where Y^i^_1_(c^i^_1_), Y^i^_2_(c^i^_2_), and Y^i^_3_(c^i^_3_) are the properties of aqueous solutions of individual compounds 1, 2, and 3 at the same concentration i for each compound (c^i^_1_ = c^i^_2_ = c^i^_3_); k_1_, k_2_, and k_3_ are the constants. It was suggested [1] that the above relationship is due to the properties of aqueous solutions that are derived from solute–water interactions.

Aqueous two-phase systems (ATPSs) are formed in aqueous mixtures of two polymers, such as dextran and poly (ethylene glycol) (PEG) or Ficoll, or in aqueous mixtures of a single polymer, such as PEG, and salt, such as sodium sulfate, phosphate or citrate, when the concentrations of the polymers/salt exceed certain threshold [2,3,4,5,6]. These systems have low interfacial tension and water constitutes up to 80–90 mol % of each phase, thereby providing benign media for biological products. The ATPSs may be used for extraction and separation of proteins, nucleic acids, viruses, cells, etc. [2,3,4,5,6]. An important fundamental advantage of ATPSs is the solvent similarity between the two phases. This similarity enables the design of ATPSs with exquisite sensitivity to very small changes in the structure of the solute. As an example, changes in the protein structure, such as a single-point mutation, glycosylation, phosphorylation, and even conformation may be easily detected by analysis of the protein partitioning in ATPS [7]. The aforementioned high sensitivity of ATPS partitioning to the protein structural changes serves as the basis for development of a new generation of clinical diagnostic tests [8].

Solute partition behavior in a given ATPS is characterized by partition coefficient, K, defined as the ratio of the solute concentration in the upper phase to that in the lower phase. It has been established that the logarithm of the partition coefficients of any solute (from small organic compound to proteins) may be described as a linear function of a sum of different solute–solvent interactions in the two phases [9,10]:
logK = S_s_Δπ* + B_s_Δα + A_s_Δβ + C_s_c,(2)
where K is the solute partition coefficient; Δπ*, Δα, Δβ, and c are the differences between the solvent properties of the top and bottom phases (solvent dipolarity/polarizability, π∗, hydrogen-bond donor acidity, α, hydrogen-bond acceptor basicity, β, and electrostatic interactions, c, respectively); and S_s_, B_s_, A_s_, and C_s_ are constants (solute-specific coefficients) that describe the complementary interactions of the solute with the solvent media in the coexisting phases; the subscript ‘s’ designates the solute.

The differences between the solvent dipolarity/polarizability, Δπ∗, hydrogen-bond donor acidity, Δα, and hydrogen-bond acceptor basicity, Δβ, may be quantified with some solvatochromic dyes [9,10]. The difference between the electrostatic properties of the phases may be determined based on the analysis of the partition coefficients of a homologous series of sodium salts of dinitrophenyl (DNP-) amino acids with aliphatic alkyl side-chains [6,9,10]. DNP amino acids contain a specific chromophore, N-2,4-dinitrophenyl, which is used as a means for the evaluation of the concentrations of these modified compounds in phases by direct optical absorbance measurements, thereby significantly increasing the accuracy of determination of their partition coefficient values. It has been shown that for a given compound (including proteins), the solute-specific coefficients may be determined by multiple linear regression analysis of the partition coefficients of the compound in multiple ATPSs with the same ionic composition. It should be mentioned that the aforementioned Equation (2) is applicable to ATPS formed by two polymers [9,10] as well as to those formed by a single polymer and a salt [11].

Often, it is important to have a possibility to manipulate partition coefficient of a given solute in the ATPS of a fixed polymer composition. If ATPS is used for extraction, an increase (decrease) of the partition coefficient of the target solute is necessary to increase the recovery of the solute. In analytical applications, it is desirable for partition coefficient of the target solute to be within a certain range, in order to ensure that concentrations of the solute can be measured reliably in both phases. There are two types of additives that can be used to manipulate solute partition behavior in ATPS. One type includes nonionic organic compounds, such as trimethylamine N-oxide (TMAO), sorbitol, and other additives capable of affecting the differences between the solvent properties of the coexisting phases, such as Δπ∗, Δα, and Δβ in Equation (1) [9,10]. The other more generally used type of additives includes various inorganic salts, such as NaCl, Na_2_SO_4_, NaClO_4_, etc. [6]. Although these additives do not affect the aforementioned solvent properties too significantly, their effects on the difference between the electrostatic properties of the phases may be very pronounced [12]. 

It was demonstrated [9,10] that the solute-specific coefficients S_s_, B_s_, A_s_, and C_s_ determined in ATPS formed by various pairs of two nonionic polymers are constant, if the ATPSs have the same ionic composition. This fact proves that solutes do not interact with the phase-forming polymers, and that partition coefficients of solutes in ATPSs are governed by the differences between the solvent properties of aqueous media in the coexisting phases. Hence, it is possible to assume that the solute partition coefficient may be viewed as a relative measure of the solute response to changes in its aqueous environment. If true, it follows that the solute partition coefficient may be considered as an important physicochemical property of a given solute.

If the solute partition coefficients in ATPSs of various ionic compositions may be considered as a physicochemical property of a given compound, it should be expected that the logarithms of partition coefficients of three different compounds are linearly interrelated according to Equation (1). In this case, however, Y^i^_1,_ Y^i^_2,_ and Y^i^_3_ are logarithms of partition coefficients of solutes 1, 2, and 3 at the i-th ionic composition of aqueous two-phase system, since the partition coefficient of each solute is independent of the solute concentration (if the conditions do not induce solute aggregation).

The purpose of this work was to explore if the relationships described by Equation (1) do exist for partition coefficients of small organic compounds and proteins in aqueous two-phase systems, for the distribution coefficients of drugs in octanol-buffer systems, and finally to examine if the same relationship may exist for the partition coefficients of drugs between blood and various tissues in rats in vivo.

## 2. Materials and Methods 

### 2.1. Materials

The data analyzed in this study and reported previously (see references in Appendix A) were obtained using the materials described below.

#### 2.1.1. Polymers

Polyethylene glycol PEG-8000 with a number average molecular weight (Mn) of 8000 Da; polyethylene glycol PEG-10000 with Mn of 10,000 Da; polyethylene glycol 6000, Mn = 6000 Da; polyethylene glycol 4000, Mn = 4000 Da; polyethylene glycol 1000, Mn = 1000 Da, and polyethylene glycol 600, Mn = 600 Da were purchased from Sigma-Aldrich (St. Louis, MO, USA) and Dextran-75 (Lot 119945) with an average molecular weight (Mw) of 75,000 Da by light scattering was purchased from USB Corporation (Cleveland, OH, USA). Ucon 50-HB-5100, Mw = 3930 Da was purchased from Dow-Chemical (Midland, MI, USA). Ficoll 70, Mw ~ 70,000 Da was purchased from GE Healthcare Biosciences AB (Sweden). All polymers were used without further purification.

#### 2.1.2. Organic Compounds 

Dinitrophenylated (DNP) amino acids—DNP-alanine, DNP-norvaline, DNP-norleucine, and DNP-α-amino-n-octanoic acid, were purchased from Sigma–Aldrich. The sodium salts of the DNP-amino acids were prepared by titration as described in [10,11,13,14,15,16,17,18,19]. Adenine, adenosine, adenosine monophosphate Na salt, adenosine diphosphate Na salt, adenosine triphosphate Na salt, 4-aminophenol, benzyl alcohol, caffeine, coumarin, methyl anthranilate, p-nitrophenyl-α-D-glucopyranoside, sorbitol, sucrose, trehalose, phenol, 2-phenylethanol, trimethylamine N-oxide (TMAO), and vanillin were purchased from Sigma-Aldrich and used without further purification as reported in [10,11,13,14,15,16,17,18,19].

#### 2.1.3. Drug Compounds

All drug compounds were purchased from Sigma-Aldrich (St. Louis, MO, USA) except atenolol which was obtained from MP Biomed (Santa Ana, CA, USA) and used as received. The purity of compounds was >95%, as specified in accompanying documentation. 1-Octanol (J.T. Baker Cat# 9085-01) was purchased from Arctic White (Bethleham, PA, USA). 

All inorganic salts and other chemicals used were of analytical-reagent grade or HPLC grade.

#### 2.1.4. Proteins

Human serum albumin (globulin and fatty acids free), bovine hemoglobin, human hemoglobin, α-chymotrypsin, α-chymotrypsinogen A from bovine pancreas, concanavalin A from Canavalia ensiformis (jack beans), cytochrome c from equine heart, β-lactoglobulin A and β-lactoglobulin B from bovine milk, ribonuclease A and ribonuclease B from bovine pancreas, subtilisin A from Bacillus licheniformis, and trypsinogen from bovine pancreas were purchased from Sigma–Aldrich. Lysozyme (salt free) from chicken egg white was obtained from Worthington Biochemical Corp. (Lakewood, NJ, USA). Porcine pancreatic lipase was purchased from USB Corp. (Solon, OH, USA). Purity of all proteins was verified by electrophoresis.

### 2.2. Methods

The relationships of the logarithms of partition coefficients for all compounds (including proteins) were examined with the software package TableCurve 3D, v.2.04 (Systat Software, Inc., San Jose, CA, USA).

## 3. Results and Discussion

### 3.1. Small Organic Compounds in Aqueous Two-Phase Systems

The partition coefficients (and their logarithms) reported for various organic compounds in different PEG-Na_2_SO_4_ ATPSs with and without various salt additives (NaCl, NaSCN, NaClO_4_, and NaH_2_PO_4_) with the concentrations varied from zero to ~1.9 M [20] and in ATPSs formed by various pairs of nonionic polymers with different ionic compositions are listed in Appendix A. Typical linear relationships observed for three different compounds, vanillin, phenol and benzyl alcohol, coumarin and methyl anthranilate, and adenine, adenosine monophosphate and adenosine diphosphate, are illustrated graphically in Figure 1A–C. The coefficients and statistical characteristics of the relationships typically observed in these analyses are listed in Table 1. 

It should be noted that different organic compounds were examined in various sets of ATPSs, and that these sets for some compounds overlap to a very limited degree. As one of the most illustrative examples, partition coefficients for benzyl alcohol under 70 different ionic compositions vary from 0.72 to 8.0, those for phenol under same conditions vary from 0.72 to 78.0, and those for vanillin—from 1.19 to 19.1. The relationship between logarithms of all these partition coefficients for the three compounds is very well described by Equation (1) (see Table 1). In one case, for the adenosine–phenol–glucoside relationship, the logarithms of the partition coefficients for adenosine do not fit the linear relationship under high concentrations of NaClO_4_ and NaH_2_PO_4_ as indicated in the footnote to Table 1. 

We can consider the change in the partition coefficient of a given compound under the varied ionic compositions of an ATPS as a measure of the response of this compound to the changes in the ATPS ionic composition. The relationships observed imply that the responses of all the organic compounds examined so far, being highly variable, appear correlated for any three different compounds. The only plausible explanation we may suggest is the previously reported [15] cooperativity of different types of polar solute–solvent interactions in aqueous media. It should be noted that all the relationships are observed for essentially nonionic compounds. If the logarithms of partition coefficients for a charged compound, such as DNP-norvaline Na, are used with those for two nonionic compounds, the linear relationship observed is much less robust, likely because the responses of a charged compound to changes in the ionic composition are different from those of nonionic compounds. With the total number of 14 organic compounds analyzed in this study, the number of the interrelationships between partition coefficients for three different compounds exceeds 360. Therefore, only some of the relationships are listed in Table 1. 

### 3.2. Proteins in Aqueous Two-Phase Systems

Partition coefficients for various proteins show that under varied ionic compositions the K-values vary quite significantly. As an example, for lysozyme, the partition coefficients vary from 0.036 to ca. 94, for α-chymotrypsinogen—from 0.0098 to 76.6, and for trypsinogen—from 0.015 to 77.5. As expected, there are multiple ionic compositions, when the partition coefficients for one or more proteins do not fit the linear relationship described by Equation (1). Analysis of the ionic compositions, under which the partition coefficients of proteins do not fit the relationship, shows that most commonly, these compositions correspond to high concentration of salt in ATPSs formed by PEG and Na_2_SO_4_ (ATPSs #23–37, Appendix A) or phosphate buffer (ATPS # 38, Appendix A) or in two-polymer ATPSs with salt additives, such as 1.05 M NaCl (ATPS # 62, Appendix A). More surprisingly, it seems to be the fact that for multitude of proteins examined here, there are many ionic compositions, where the proteins’ responses to their environment are correlated with the responses of small organic compounds. Several proteins were examined under ionic composition conditions used for studying small organic compounds. Analysis of these data from Appendix A showed that there are several sets of linear relationships for two small compounds and one protein. Characteristics of these relationships are provided in Table 2. The aforementioned data imply that the similar forces are driving partition behavior of small compounds and proteins. It also follows from these observations that the linear relationships described by Equation (1) are typical for compounds in aqueous media.

The partition coefficients (and their logarithms) for proteins reported previously in various polymer–polymer and PEG–Na_2_SO_4_ ATPSs with various salt additives are listed in Appendix A. For the total number of 15 proteins analyzed here, the overall number of the interrelationships between partition coefficients for three different proteins exceeds 450. Therefore, only some of the relationships are listed in Table 2. Analysis of the data for sets of three different proteins shows that Equation (1) holds for proteins. Typical relationships observed for various proteins, such as α-chymotrypsin (CHY), β-lactoglobulin A (bLGA), ribonuclease A (RNase A), lysozyme (HEL), and ribonuclease B (RNase B), are illustrated graphically in Figure 2A,B. Typical relationship between two small organic compounds and protein, benzyl alcohol, vanillin, and α-chymotrypsinogen (CHTG), is illustrated graphically in Figure 2C. The coefficients and statistical characteristics of the typical relationships observed are listed in Table 2. The ATPS compositions, under which each relationship does not hold are listed in Table 3 as the ID numbers of ATPSs. These ID numbers correspond to the compositions of the ATPSs listed in Appendix A.

### 3.3. Drugs in Octanol-Buffer Systems

One of the important characteristics of a compound is its lipophilicity (which represents a measure of the tendency of a compound to move from the aqueous phase into lipids) that can be evaluated based on the partition of this compound in the octanol-water systems. In this case, lipophilicity of a given solute/compound is estimated based on the octanol-buffer partition coefficient measured as the ratio of the solute concentration in the organic phase to that in the aqueous phase. Analysis of distribution coefficients of drugs in octanol-buffer systems with different ionic composition reported in [21] and listed in Appendix A showed that the relationships described by Equation (1) hold for a very limited sets of compounds examined (see in Table 3). Three examples of such relationships for various sets of drugs are graphically illustrated in Figure 3A–C. There are multiple factors affecting the distribution of compounds in octanol-buffer system. As an example, the ionic composition of an aqueous phase may affect the octanol solubility in the phase, and the interactions of some drugs with octanol may differ significantly. 

In any case, the changes in solute distribution in octanol-buffer system under varied ionic composition may hardly be considered in terms of the compound response to ionic composition only. The detailed analysis of compounds fitting the relationships according Equation (1) and those not fitting it is beyond the scope of the present study. Partition coefficients in octanol-buffer systems for various drug compounds were compared to verify the hypothesis that the linear relationship under discussion is valid mostly for aqueous media. It seems reasonable to suggest that only compounds with relatively similar energies of solute–solvent interactions with octanol may display the relationship described by Equation (1). The data presented here confirm our hypothesis that the Equation (1) is valid mostly for compounds in aqueous media.

### 3.4. Drugs Distribution between Blood and Other Tissues In Vivo

As suggested above, the linear relationships described by Equation (1) hold for compounds in aqueous media. If this assumption is true, it should be expected that the relationships described by Equation (1) should be observed for the drug blood–tissue partition coefficients as well. We explored this issue using the data reported in [22] for in vivo rat blood–tissue distribution of multiple drugs. The data are presented in Appendix A in the convenient format. The examples of relationships observed for the logarithms of the partition coefficients of three different drugs are listed in Table 4, and two illustrative examples of the relationships observed for pindolol, metoprolol, and oxprenolol and for lomefloxacin, nalidixic acid, and ofloxacin are graphically presented in Figure 4A,B. These examples not only confirm the above conclusion but also suggest a convenient novel approach to the analysis of possible side-effects of drug candidates, since it provides a very simple route for comparison of drug candidates on the stage of testing in animals. The tissues for which the blood–tissue partition coefficients do not fit the relationship described by Equation (1) may be considered as those, where the compounds may be engaged in specific or non-specific interactions with some components of the tissue.

## 4. Conclusions

Analysis of the previously reported partition coefficients of small organic compounds and proteins in different aqueous two-phase systems under varied ionic compositions shows that the linear relationship between logarithms of partition coefficients for three solutes holds for all nonionic organic compounds under essentially all ionic compositions. For proteins it also hold though there are ionic compositions under which partition coefficients for one or more proteins in the set considered may not fit the linear relationship. 

It was suggested that the linear relationship under consideration is valid mostly for conditions when water is the solvent in the two coexisting phases. This assumption was confirmed by results of analysis of distribution coefficients for drugs in octanol-buffer systems with varied ionic composition of the buffer indicating that the relationship holds only for the limited number of drugs (17 out of 28) and drugs combinations.

Based on the above assumption that the linear relationship under consideration is valid for aqueous media analysis of the partition coefficients for drugs between blood and various tissues in rats in vivo reported in the literature was performed. The results of this analysis not only confirm the assumption but enable one to detect the tissues with components of which the drug(s) may be engaged in direct interactions.

## Figures and Tables

**Figure 1 polymers-12-01452-f001:**
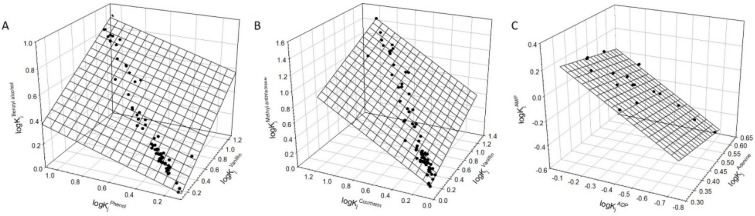
(**A**) Linear relationship between logarithms of partition coefficients of vanillin, phenol, and benzyl alcohol in various aqueous two-phase systems (Data from Appendix A). The plane corresponds to Equation (1). Error bars are the same size as/or smaller than the symbols. (**B**). Linear relationship between logarithms of partition coefficients of vanillin, coumarin, and methyl anthranilate in various aqueous two-phase systems (Data from Appendix A). The plane corresponds to Equation (1). Error bars are the same size as/or smaller than the symbols. (**C**). Linear relationship between logarithms of partition coefficients of adenine, adenosine diphosphate (ADP), and adenosine monophosphate (AMP) in various aqueous two-phase systems (Data from Appendix A). The plane corresponds to Equation (1). Error bars are the same size as/or smaller than the symbols.

**Figure 2 polymers-12-01452-f002:**
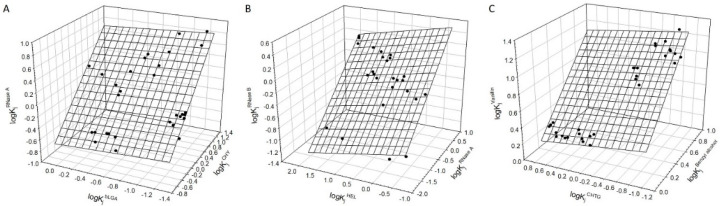
(**A**) Linear relationship between logarithms of partition coefficients of α-chymotrypsin (CHY), β-lactoglobulin A (bLGA), and ribonuclease A (RNase A) in various aqueous two-phase systems (Data from Appendix A). The plane corresponds to Equation (1). Error bars are the same size as/or smaller than the symbols. (**B**) Linear relationship between logarithms of partition coefficients of ribonuclease A (RNase A), lysozyme (HEL), and ribonuclease B (RNase B) in various aqueous two-phase systems (Data from Appendix A). The plane corresponds to Equation (1). Error bars are the same size as/or smaller than the symbols. (**C**)**.** Linear relationship between logarithms of partition coefficients of benzyl alcohol, α-chymotrypsinogen (CHTG), and vanillin in various aqueous two-phase systems (Data from Appendix A). The plane corresponds to Equation (1). Error bars are the same size as/or smaller than the symbols.

**Figure 3 polymers-12-01452-f003:**
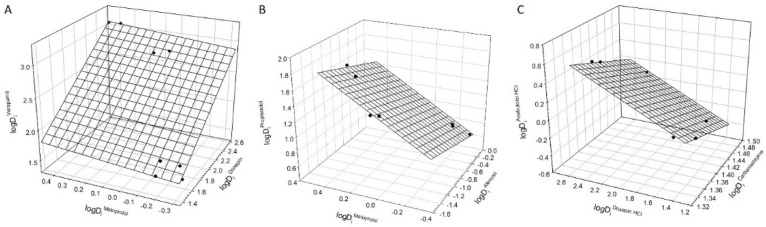
(**A**) Linear relationship between logarithms of distribution coefficients of doxepin, metoprolol, and verapamil in octanol-buffer, pH 7.4 with various ionic compositions of the buffer (Data from Appendix A). The plane corresponds to Equation (1). Error bars are the same size as/or smaller than the symbols. (**B**) Linear relationship between logarithms of distribution coefficients of atenolol, metoprolol, and propranolol in octanol-buffer, pH 7.4 with various ionic compositions of the buffer (Data from Appendix A). The plane corresponds to Equation (1). Error bars are the same size as/or smaller than the symbols. (**C**) Linear relationship between logarithms of distribution coefficients of carbamazepine, doxepin HCl, and acebutolol HCl in octanol-buffer, pH 7.4 with various ionic compositions of the buffer (Data from Appendix A). The plane corresponds to Equation (1). Error bars are the same size as/or smaller than the symbols.

**Figure 4 polymers-12-01452-f004:**
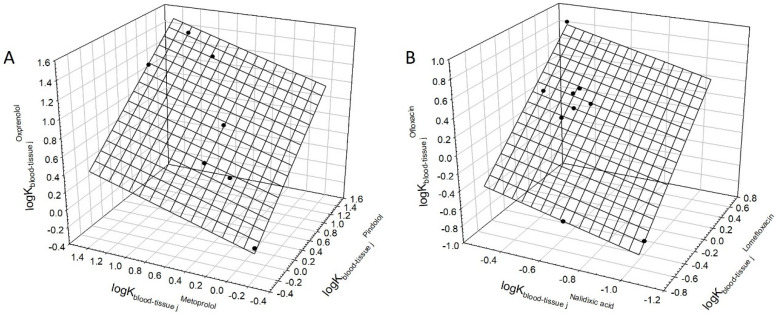
(**A**) Linear relationship between logarithms of partition coefficients of pindolol, metoprolol, and oxprenolol between blood and various tissues in rats in vivo (data from [22], see in Appendix A). The plane corresponds to Equation (1). Error bars are the same size as/or smaller than the symbols. (**B**) Linear relationship between logarithms of partition coefficients of lomefloxacin, nalidixic acid, and ofloxacin between blood and various tissues in rats in vivo (data from [22], see in Appendix A). The plane corresponds to Equation (1). Error bars are the same size as/or smaller than the symbols.

**Table 1 polymers-12-01452-t001:** Coefficients and statistical characteristics of linear relationships (k_1_, k_2_, and k_3_ are the constants in equation (1); N is the number of solutes examined; r^2^ is the correlation coefficient; SD is the standard deviation; and F is the ratio of variance) between logarithms of partition coefficients for various organic compounds in aqueous two-phase systems (ATPSs) of different ionic compositions (for raw data see in Appendix A).

logK (X).	logK (Y)	logK (Z)	k_1_	k_2_	k_3_	N	r^2^	SD	F
Caffeine	Phenol	Glucoside ^a^	−0.039_0.009_	0.65_0.05_	0.31_0.02_	63	0.9802	0.034	1488
Vanillin	Phenol	Benzyl alcohol	0	0.43_0.06_	0.31_0.07_	70	0.9841	0.030	2069
Vanillin	Coumarin	Benzyl alcohol	0.021_0.006_	0.40_0.04_	0.33_0.05_	70	0.9883	0.025	2818
Vanillin	Coumarin	Methyl anthranilate	0.023_0.008_	0.44_0.06_	0.75_0.07_	70	0.9902	0.037	3395
Vanillin	Glucoside ^a^	Coumarin	0	0.44_0.07_	0.8_0.1_	70	0.9726	0.052	1187
Vanillin	Glucoside ^a^	2-Phenylethanol	0	0.55_0.06_	0.5_0.1_	70	0.9713	0.046	1133
Phenol	Methyl anthranilate	Benzyl alcohol	0	0.34_0.07_	0.36_0.06_	70	0.9823	0.031	1859
Benzyl alcohol	Caffeine	Coumarin	0	0.8_0.1_	0.8_0.2_	31	0.9845	0.045	888
Glucoside ^a^	Coumarin	Caffeine	−0.03_0.01_	0.3_0.1_	0.31_0.06_	31	0.9772	0.027	600
Adenine	ADP	AMP	0.11_0.01_	−0.02_0.002_	0.87_0.03_	18	0.9888	0.019	661
Glucoside ^a^	Adenine	Caffeine	−0.030_0.008_	0.79_0.03_	0.08_0.02_	23	0.9890	0.014	901
AMP ^b^	ATP ^b^	ADP ^b^	−0.06_0.02_	0.62_0.06_	0.39_0.05_	18	0.9904	0.019	771
Vanillin	Caffeine	Glucoside ^a^	0	0.27_0.06_	0.6_0.1_	31	0.9775	0.032	607
2-Phenylethanol	Coumarin	Benzyl alcohol	0.028_0.006_	0.51_0.07_	0.31_0.06_	70	0.9849	0.029	2189
Phenol	AMP ^b^	Glucoside ^a^	−0.13_0.02_	0.66_0.02_	0.12_0.02_	18	0.9834	0.010	443
Adenosine	Phenol	Glucoside ^a^	0	0.25_0.03_	0.36_0.02_	29 ^c^	0.9889	0.015	1163
Methyl anthranilate	Caffeine	Glucoside ^a^	0	0.30_0.07_	0.5_0.2_	31	0.9734	0.035	512

^a^ 4-nitrophenol-α-D-glucopyranoside; ^b^ AMP—adenosine monophosphate, ADP—adenosine diphosphate, ATP—adenosine triphosphate; ^c^ 1.759 M NaClO_4_, 0.556 M–1.751 M NaH_2_PO_4_ (all indicated salt concentrations correspond to conditions under which the partition coefficients for the compounds do not fit the linear relationship indicated).

**Table 2 polymers-12-01452-t002:** Coefficients and statistical characteristics of linear relationships between logarithms of partition coefficients for various proteins * (see Appendix A) and between logarithms of partition coefficients for two small compounds and one protein in ATPS of different polymer and ionic compositions (data see Appendix A).

logK-X	logK-Y	logK-Z	k_1_	k_2_	k_3_	N	r^2^	SD	F	Conditions ^a^
CHY	RNase B	RNase A	−0.23_0.02_	0.94_0.04_	−0.25_0.07_	30	0.9840	0.056	831	23–28,39,42,60,63
RNase B	CHTG	CHY	−0.11_0.03_	0.85_0.05_	0.58_0.03_	22	0.9882	0.050	792	23–27,29–32,36,39–41,63,64,71,75
RNase A	BHb	CHY	0.28_0.01_	0.66_0.04_	0.24_0.03_	23	0.9925	0.058	1325	63
BHb	CHTG	CHY	0	0.49_0.04_	0.30_0.04_	19	0.9873	0.062	623	27,36,63
CHY	bLGA	ConA	−0.50_0.02_	0.11_0.03_	0.27_0.02_	27	0.9232	0.059	144	29,30,32–35,38–41,47,60
HHb	TRY	BHb	−0.31_0.02_	0.79_0.03_	0.14_0.04_	24	0.9891	0.068	950	21,27,33
bLGB	HEL	bLGA	0	1.10_0.04_	−0.18_0.01_	20	0.9876	0.045	679	23–27,35,38,60–67,70,72-75
Lipase	CHTG	CHY	0	2.7_0.7_	0.77_0.08_	13	0.9910	0.067	551	4,8,9,11,12,16,57,58
CHY	bLGA	RNase A	−0.25_0.02_	0.85_0.02_	−0.09_0.02_	30	0.9870	0.050	1025	23–28,38–40,60,63
HSA	bLGA	RNase A	−0.18_0.04_	0.71_0.04_	−0.81_0.09_	10	0.9781	0.057	156	28,31,34,35,37
Sub A	RNase A	RNase B	0	−0.8_0.2_	0.58_0.02_	19	0.9803	0.054	398	25,26,28,33,37,38,60,62
RNase A	HEL	RNase B	−0.09_0.02_	0.58_0.02_	−0.13_0.02_	27	0.9676	0.065	359	25,38–47,49,52,64,68,70–75
HHb	CHTG	RNase A	−0.28_0.02_	−0.20_0.04_	1.42_0.06_	21	0.9888	0.070	797	24,25,30,60,62,63,66
bLGB	HHb	ConA	−0.31_0.04_	0.54_0.05_	0.14_0.02_	22	0.9622	0.051	242	60,62,72–75
CHY	SubA	ConA	−0.91_0.03_	0.40_0.02_	0.6_0.1_	19	0.9709	0.050	267	23,30,32,37,38,62
CHTG	ConA	Lipase	−0.13_0.01_	0.08_0.01_	0.10_0.02_	21	0.8672	0.029	58.8	-
TRY	RNase B	RNase A	0	0.51_0.04_	0.61_0.09_	32	0.9787	0.081	667	23,24,28,36,60,71,73
HEL	TRY	CHTG	0.39_0.02_	0.15_0.02_	0.58_0.03_	25	0.9678	0.073	316	26,32–37,50–52,62,66,73,74
bLGA	RNasa A	bLGB	−0.25_0.02_	0.50_0.02_	0.08_0.02_	25	0.9884	0.046	934	23,39–41,45–47,60,62,65–67,72–75
Benzyl alcohol	CHTG	Vanillin	0.05_0.02_	1.25_0.04_	−0.11_0.02_	31	0.9920	0.036	1745	-
Benzyl alcohol	HEL	Vanillin	−0.03_0.02_	1.46_0.03_	−0.16_0.008_	30	0.9898	0.041	1313	15
Caffeine	CHY	Glucoside ^b^	004_0.01_	0.97_0.08_	−0.06_0.02_	31	0.9664	0.040	403	-
Caffeine	Phenol	Lipase	−0.06_0.01_	−0.24_0.07_	−0.12_0.04_	25	0.9501	0.019	209	14

* Proteins: α-Chymotrypsinogen—CHTG; Chymotrypsin—CHY; Concanavalin A—ConA; Cytochrome c—CytC; Hemoglobin bovine—BHb; Hemoglobin human—HHb; β-Lactoglobulin A—bLGA; β-Lactoglobulin B—bLGB; Lysozyme—HEL; Subtilisin A—SubA; Trypsinogen—TRY; Ribonuclease A—RNase A; Ribonuclease B—RNase B. ^a^ ATPS in which the partition coefficients for the indicated proteins do not fit the linear relationships described by Equation (1) (the list of ATPSs see in Appendix A for small compounds). ^b^ Glucoside - 4-nitrophenol-α-D-glucopyranoside.

**Table 3 polymers-12-01452-t003:** Coefficients and statistical characteristics of linear relationships between logarithms of partition coefficients for various organic compounds in octanol-buffer systems of different ionic compositions (data from [21], see Appendix A).

logK-X	logK-Y	logK-Z	k_1_	k_2_	k_3_	N	r^2^	SD	F
Terbutaline	Piroxicam	Clonidine HCl	0.8_0.1_	0.27_0.09_	−0.9_0.06_	7 ^a^	0.9888	0.038	176
Atenolol	Metoprolol (1/2 tartrate)	Propranolol	0.97_0.04_	−0.24_0.05_	1.02_0.08_	7 ^b^	0.9896	0.028	191
Acebutolol HCl	Desipramine HCl	Metoprolol (1/2 tartrate)	−0.60_0.07_	0.70_0.09_	0.45_0.06_	8	0.9941	0.028	419
Atenolol	Desipramine HCl	Propranolol	0	−0.13_0.05_	0.99_0.09_	8	0.9850	0.057	165
Minaprine 2HCl	Mefexamide HCl	Verapamil	1.4_0.1_	0.31_0.09_	0.9_0.3_	8	0.9782	0.11	112
Furosemide	Doxepin HCl	Diclofenac Na	2.9_0.2_	1.8_0.1_	0.2_0.03_	8	0.9852	0.047	167
Doxepin HCl	Metoprolol (1/2 tartrate)	Verapamil	0.39_0.08_	0.98_0.04_	0.22_0.08_	8	0.9988	0.026	2128
Atenolol	Acebutolol HCl	Propranolol	0.4_0.2_	1.9_0.4_	−1.1_0.2_	8	0.9656	0.23	70
Verapamil	Acebutolol HCl	3-Hydroxytryptophan	−2.34_0.05_	0.14_0.02_	−0.39_0.05_	7 ^b^	0.9553	0.015	32
Chlorpromazine	Propranolol	Verapamil	0	0.67_0.06_	0.5_0.1_	6 ^b,c^	0.9996	0.016	3647
Carbamazepine	Doxepin HCl	Acebutolol HCl	2.5_0.3_	−2.4_0.2_	0.47_0.02_	6 ^a,d^	0.9938	0.027	241

^a^ 0.01 M NaPB; ^b^ 0.15 M NaCl in 0.01 M NaPB; ^c^ 0.15 M NaCl in 0.10 M NaPB; ^d^ 0.10 M NaPB (NaPB—sodium phosphate buffer); all the buffer composition indicated correspond to conditions under which the distribution coefficients for the indicated compounds do not fit the linear relationships.

**Table 4 polymers-12-01452-t004:** Coefficients and statistical characteristics of linear relationships (Equation (1)) between logarithms of partition coefficients for various drugs between blood and different tissues * in rats in vivo (data from [22], see Appendix A).

logK-X	logK-Y	logK-Z	k_1_	k_2_	k_3_	N	r^2^	SD	F	Tissues ^a^
Thiopental	Tenoxicam	Salicylic acid	−0.33_0.08_	0.70_0.07_	0.28_0.07_	7	0.9947	0.088	372	Liver, skin
Pindolol	Metoprolol	Oxprenolol	0	0.68_0.09_	0.36_0.09_	7	0.9886	0.062	130	Brain, heart, kidney
Imipramine	Diazepam	Fingolimod	0.48_0.03_	0.61_0.04_	0.32_0.05_	5	0.9998	0.025	4573	Brain, heart
Acebutolol	Ftorafur	Bisoprolol	0.3_0.1_	0.58_0.07_	0.8_0.2_	5	0.9885	0.088	86.2	Brain, liver, lungs, skin
Ftorafur	Fentanyl	Barbital	0	0.37_0.09_	−0.10_0.03_	8	0.9469	0.030	53.5	Kidney, liver, lungs
Ceftazidime	Bisoprolol	Fentanyl	0.32_0.05_	0.13_0.04_	0.60_0.04_	5	0.9944	0.038	178	Adipose, heart, liver
Fentanyl	Barbital	Acebutolol	05_0.1_	0.5_0.1_	1.8_0.2_	6	0.9693	0.096	47.3	Adipose, brain, intestine
Acebutolol	Propranolol	Metoprolol	0	0.18_0.07_	0.60_0.06_	5	0.9877	0.077	120	Adipose, intestine, kidney, liver
Phenytoin	Phencyclidine	Pentazocine	0.54_0.06_	0.32_0.08_	0.50_0.07_	5	0.9985	0.075	660	Adipose, brain, liver
Thiopental	Tenxicam	Timolol	1.15_0.03_	−0.58_0.04_	0.61_0.02_	5	0.9970	0.037	331	Intestine, kidney, lungs, skin
Tolbutamide	Triazolam	Valproic acid	0	1.97_0.12_	1.7_0.1_	4	0.9969	0.027	161	Kidney, liver
Quinidine	Salicylic acid	Thiopental	0.45_0.01_	0.04_0.01_	0.83_0.01_	4	0.9998	0.004	2313	Brain, liver, muscle
Lomefloxacin	Nalidixic acid	Ofloxacin	0.33_0.05_	0.88_0.04_	0.52_0.08_	9	0.9969	0.031	959	Liver
Barbital	Alprazolam	PEB acid ^b^	0.44_0.09_	1.6_0.3_	−0.8_0.2_	7	0.9509	0.055	38.7	-
Betaxolol	Ceftazidime	Bisoprolol	0	0.82_0.05_	0.18_0.06_	7	0.9905	0.071	209	Intestine
Cotinine	Ceftazidime	Cefazolin	0.18_0.01_	0.94_0.07_	0.95_0.01_	5	0.9999	0.009	10617	Liver
Midazolam	Metoprolol	Lomefloxacin	0	−0.46_0.08_	0.64_0.05_	6	0.9842	0.055	93.2	Brain, kidney, lungs
Nalidixic acid	Nicotine	Oxrenolol	0.9_0.1_	0.8_0.1_	0.53_0.08_	5	0.9936	0.061	155	Brain, intestine, lungs, skin
Pindolol	Oxprenolol	Phenytoin	0	−1.03_0.08_	1.15_0.08_	6	0.9859	0.030	105	Kidney, lungs, muscle
Matrine	Midazolam	Metuprolol	0	0.71_0.07_	0.68_0.08_	4	0.9961	0.039	129	Adipose, brain, heart, muscle

* Tissues where the drug concentration was measured are different for each drug (see in Appendix A). ^a^ Tissues indicated are those for which logK blood–tissue does not fit the linear relationship (suggested explanation see in text); ^b^ PEB acid—5-propyl-5-ethyl barbituric acid.

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
