# Peer review of "Linear Relationships between Partition Coefficients of Different Organic Compounds and Proteins in Aqueous Two-Phase Systems of Various Polymer and Ionic Compositions"

_polymers, 2020, doi:10.3390/polym12071452_

Round 1

Reviewer 1 Report

The article can be accepted for publication after minor adjustments and clarifications.

  1. Why were octanol-buffer systems used?
  2. Why were Dinitrophenylated (DNP) amino acids used? Why weren't pure amino acids used?
  3. Adjust all units of measurement in the text, for example for concentration M was used, but mol / L must be used.
  4. In the tables presented, there are some parameters that have not been defined.
  5.  

Author Response

The article can be accepted for publication after minor adjustments and clarifications.

REPLY: We are thankful to this reviewer for useful suggestions. We incorporated all suggested clarifications and changes to the revised manuscript.

1) Why were octanol-buffer systems used?

REPLY: One of the important characteristics of a compound is its lipophilicity (which represents a measure of the tendency of a compound to move from the aqueous phase into lipids) that can be evaluated based on the partition of this compound in the octanol-water systems. In this case, lipophilicity of a given solute/compound is estimated based on the octanol-buffer partition coefficient measured as the ratio of the solute concentration in the organic phase to that in the aqueous phase. Partition coefficients in octanol-buffer systems for various drug compounds were compared to verify the hypothesis that the linear relationship under discussion is valid mostly for aqueous media. It seems reasonable to suggest that only compounds with relatively similar energies of solute-solvent interactions with octanol may display the relationship described by Eq. 1. The data presented here confirm our hypothesis that the Eq.1 is valid mostly for compounds in aqueous media. The corresponding clarifications were added to the revised manuscript.

2) Why were Dinitrophenylated (DNP) amino acids used? Why weren't pure amino acids used?

REPLY: DNP amino acids contain a specific chromophore, N-2,4-dinitrophenyl, which is used as a means for the evaluation of the concentrations of these modified compounds in phases by direct optical absorbance measurements, thereby significantly increasing the accuracy of determination of their partition coefficient values. The corresponding clarifications were added to the revised manuscript.

3) Adjust all units of measurement in the text, for example for concentration M was used, but mol / L must be used.

REPLY: Using M as a unit for concentration is an accepted practice in the field.  

4) In the tables presented, there are some parameters that have not been defined.

REPLY:  Thank you for pointing this out. We added the corresponding clarifications to Table 1, which reads as (k1, k2, and k3 are the constants in equation (1); N is the number of solutes examined; r2 is the correlation coefficient; SD is the standard deviation; and F is the ratio of variance)

Reviewer 2 Report

The article “Linear Relationships between Partition Coefficients of Different Organic Compounds and Proteins in Aqueous Two-Phase Systems of Various Polymer and Ionic Compositions” by Zaslavsky and colleagues analyzes the partition coefficients of small organic compounds and proteins in different aqueous two-phase systems under widely varied ionic compositions and compare them to already reported values in the literature obtained from in vivo experiments. The article is well-written and organized and the introduction and the materials and methods are described satisfactorily. In my opinion more emphasis should have been given to the comparison with the in vivo data in order to obtain more significant data with potential application to a wider audience. Similarly, a more detailed discussion trying to explain why some drugs displayed different calculated partition coefficient from the real one would have enriched the quality of the article. Nevertheless, it still is quite interesting and in my opinion it deserves publication in Polymers.

Author Response

The article “Linear Relationships between Partition Coefficients of Different Organic Compounds and Proteins in Aqueous Two-Phase Systems of Various Polymer and Ionic Compositions” by Zaslavsky and colleagues analyzes the partition coefficients of small organic compounds and proteins in different aqueous two-phase systems under widely varied ionic compositions and compare them to already reported values in the literature obtained from in vivo experiments. The article is well-written and organized and the introduction and the materials and methods are described satisfactorily. In my opinion more emphasis should have been given to the comparison with the in vivo data in order to obtain more significant data with potential application to a wider audience. Similarly, a more detailed discussion trying to explain why some drugs displayed different calculated partition coefficient from the real one would have enriched the quality of the article.

Nevertheless, it still is quite interesting and in my opinion it deserves publication in Polymers.

REPLY: We are thankful to this reviewer for high evaluation of our work. However, we are not sure that we can introduce the requested changes (see below).

In my opinion more emphasis should have been given to the comparison with the in vivo data in order to obtain more significant data with potential application to a wider audience.

REPLY: We think that at the current stage, making more emphasis on the comparison of the results of our analyses with the in vivo data would be on overinterpretation of the results presented in this study. This is definitely a very interesting question that requires additional analyses, and the corresponding work is currently in progress.

Similarly, a more detailed discussion trying to explain why some drugs displayed different calculated partition coefficient from the real one would have enriched the quality of the article.

REPLY: In our work, we used exclusively empirical partition coefficient values, and did not have any calculated partition coefficients. Therefore, we cannot address a question of why some drugs displayed different calculated partition coefficient from the real one.